# Metavirome Profiling and Dynamics of the DNA Viral Community in Seawater in Chuuk State, Federated States of Micronesia

**DOI:** 10.3390/v15061293

**Published:** 2023-05-31

**Authors:** Seung Won Jung, Kang Eun Kim, Hyun-Jung Kim, Taek-Kyun Lee

**Affiliations:** 1Library of Marine Samples, Korea Institute of Ocean Science and Technology, Geoje 53201, Republic of Korea; rkddmssl@kiost.ac.kr (K.E.K.); hjkim8845@kiost.ac.kr (H.-J.K.); 2Department of Ocean Science, University of Science and Technology, Daejeon 34113, Republic of Korea; tklee@kiost.ac.kr; 3Risk Assessment Research Center, Korea Institute of Ocean Science and Technology, Geoje 53201, Republic of Korea

**Keywords:** metavirome, DNA virus community, cyanophage, eukaryotic virus, Micronesia coastal waters

## Abstract

Despite their abundance and ecological importance, little is known about the diversity of marine viruses, in part because most cannot be cultured in the laboratory. Here, we used high-throughput viral metagenomics of uncultivated viruses to investigate the dynamics of DNA viruses in tropical seawater sampled from Chuuk State, Federated States of Micronesia, in March, June, and December 2014. Among the identified viruses, 71–79% were bacteriophages belonging to the families *Myoviridae*, *Siphoviridae*, and *Podoviridae* (*Caudoviriales*), listed in order of abundance at all sampling times. Although the measured environmental factors (temperature, salinity, and pH) remained unchanged in the seawater over time, viral dynamics changed. The proportion of cyanophages (34.7%) was highest in June, whereas the proportion of mimiviruses, phycodnaviruses, and other nucleo-cytoplasmic large DNA viruses (NCLDVs) was higher in March and December. Although host species were not analysed, the dramatic viral community change observed in June was likely due to changes in the abundance of cyanophage-infected cyanobacteria, whereas that in NCLDVs was likely due to the abundance of potential eukaryote-infected hosts. These results serve as a basis for comparative analyses of other marine viral communities, and guide policy-making when considering marine life care in Chuuk State.

## 1. Introduction

Viruses are present everywhere, and represent the most abundant and diverse organisms in the ocean, estimated to be over 10^30^. They are also the most abundant biotic components in marine ecosystems, and integral components of marine microbial loops, where they play an important role in ecosystem functioning, such as in the supplementation of dissolved organic matter [1,2,3]. The viral community is divided into RNA and DNA viruses, based on their genetic makeup. The DNA genome of a DNA virus is replicated using DNA polymerase. DNA viruses are ubiquitous, especially in marine environments where they form an important part of marine ecosystems and infect both prokaryotes and eukaryotes [4]. Bacteriophages are prokaryotic viruses that infect both eubacteria and archaea. Marine cyanobacteria, including those in the genera *Synechococcus* and *Prochlorococcus*, are abundant primary producers in the open oceans [5]. With abundant cyanobacteria, cyanophages are also highly abundant [6,7] and influence multiple primary production processes of cyanobacteria, as well as cyanobacterial ecology [8].

Most marine viruses cannot be cultured in a laboratory, which impedes research on their diversity and ecological niche in the natural environment. In addition, the small size of the virus particle poses an important barrier to microscopic studies of ecological diversity. Finally, viruses do not harbour universal marker genes [9,10]; thus, despite their abundance and ecological importance, little is known about viral diversity or viral ecology. Viral metagenomic sequencing is a powerful tool for the high-throughput analysis of uncultivatable viruses [11,12]. This tool can be used to increase the discovery of marine viral genomes, as well as the resolution of marine viral biodiversity [13]. In addition, Flaviani et al. [14] suggested that it is possible to analyse aquatic microbial diversity using viral metagenomics with even smaller sample volumes. Viral metagenomic studies performed over the past decade have revealed staggering levels of diversity and the influence of viruses on host physiology and evolution [15,16].

In the present study, to understand viral diversity in the tropical ocean, we examined the metavirome in coastal waters of Chuuk State, Federated States of Micronesia (FSM), in samples collected in March, June, and December of 2014. Chuuk State has low marine pollution, owing to its relatively small human population, and is a habitat for many coral communities and various marine microorganisms. The results of this study provide a more complete understanding of the changes in viral communities in tropical oceans over time, which can be used in further studies on potential viral infections that can threaten marine ecosystems. This study is the first comprehensive investigation of seasonal changes in viral populations in a marine environment near Chuuk State, FSM.

## 2. Materials and Methods

### Metavirome Analysis of dsDNA Viruses

To analyse the community of DNA viruses, 40 L of seawater was collected from the surface layer (1 m below the sea surface) of coastal waters using a Niskin water sampler on 3 March, 11 June, and 1 December 2014. The samples were collected at the climate change monitoring site (seawater and coral sampling point 7°45′32″ N and 151°89′25″ E) of the Korean South Pacific Ocean Research Centre (KSORC, located in Weno Island, Chuuk State), Korea Institute of Ocean Science and Technology (KIOST) Federated States of Micronesia (Figure 1).

The temperature, salinity, and pH of the Chuuk coastal water were measured at the same time as the sample collection using an environmental profiler (YSI 6600 ver. 2, YSI Inc. OH, USA). Seawater samples were placed in sterile polypropylene bottles and stored at approximately 4 °C for transportation to the KIOST KSORC. DNA viral communities were analysed according to previously described protocols [17,18], with certain modifications. To extract genomic DNA (gDNA) from the DNA viral community, 40 L of seawater was passed through a 3 μm filter (TSTP04700, 47 mm; MilliporeSigma, Burlington, MA, USA) to remove inorganic and organic particles. DNA viruses were harvested from the prefiltered seawater using flocculation, filtration, and resuspension, using FeCl_3_ as previously described [19]. To aggregate the DNA viruses onto Fe^3+^ ions, a cylindrical, black-coloured acrylic reservoir was maintained at 20 °C for 1 h. Seawater in the reservoir was mixed with three impellers at 10 rpm. The aggregated DNA viruses were harvested using a 0.2 μm polycarbonate membrane (111106; 47 mm; Whatman, Buckinghamshire, UK). Subsequently, the membrane was cut into eight pieces and placed in a 50 mL conical tube with 10 mL of suspension buffer (0.1 M EDTA, 0.2 M MgCl_2_, 0.2 M ascorbate). The flocculated viruses attached to the membrane were suspended and mixed in the buffer (pH 6, adjusted using ~5 mL 10 M NaOH). Total gDNA was extracted using the Viral Gene-spin Viral DNA/RNA Extraction Kit (iNtRON Biotechnology, Seoul, South Korea). A gDNA library was prepared using the NEBNext Ultra II DNA Library Prep Kit (Illumina Inc., San Diego, CA, USA). The sequencing library was prepared through random fragmentation of the DNA sample, followed by 5′ and 3′ adapter ligation. The adapter-ligated fragments were amplified using polymerase chain reaction (PCR) and the following conditions: denaturation at 98 °C for 30 s, followed by 12 cycles of annealing at 98 °C for 10 s and extension at 65 °C for 75 s, and a final extension cycle of 65 °C for 5 min. An Illumina-tagged universal primer (5′-AAT GAT ACG GCG ACC ACC GAG ATC TAC ACT CTT TCC CTA CAC GAC GCT CTT CCG ATC-3′) and indexed primers, designed using NEBNext Multiplex Oligos (Illumina), were used. The DNA library was subjected to paired-end sequencing using an Illumina HiSeq 2500 platform.

We followed the classification system for bacteriophages proposed by the International Committee on Taxonomy of Viruses (ICTV) for the classification system of viruses and added the new classification system proposed in 2023 by ICTV in parentheses [20,21]. Modified bioinformatics analysis was performed according to the protocol described by Kim et al. [18]. Briefly, raw sequencing data (FASTQ file) were sorted by trimming using the Qiagen CLC genomics workbench (v. 20.0.4; Qiagen, Hilden, Germany), which removes low-quality adapters based on the criteria of two ambiguous nucleotides and a minimum length of 50 nucleotides. Next, assembly analysis was performed using metaSPAdes v. 3.13.0 [22], and five k-mer sizes (k-mer: 21, 33, 55, 77, 99, and 127) were selected. Quality checks of viral contigs were performed using CheckV v.1.0, to remove contiguous host and non-viral regions from the assembled proviruses [23]. Through the CheckV quality check, only virus contigs of higher completeness (over >1000 bp) were selected and read mapping was applied using Bowtie 2 (v2.5.1) with the ‘very-sensitive-local’ parameter and 95% minimum alignment identity [24]. The virus contigs were quality checked using CheckV, and read mapping was subjected to virus taxonomy analysis using the Basic Local Alignment Search Tool (BLASTn). BLASTn was performed using the Microbial Genomic Module in the CLC Genomics Workbench setting, with the following options: NCBI Viral RefSeq database. The results were sorted according to the viral taxon (eukaryotic viruses and bacteriophages) and nucleo-cytoplasmic large DNA viruses (NCLDVs) in eukaryotic viral communities using customised taxonomic profiling assignment coding developed by Kim et al. [17] in the open-source software R within R studio (version 3.6.3) [25].

A hierarchical agglomerative clustering analysis was performed using the group average of the viral contigs and the Bray–Curtis dissimilarity method. The heatmap was plotted using the ‘ggplot2’ package [26] with a combination of custom R and R Studio (v. 1.2.5042). Alpha diversity (including Chao1, Shannon, and Simpson diversity metrics) was analysed using the vegan package in R [27].

## 3. Results and Discussion

Over the months examined, the study site seawater temperature ranged from 28.6 to 29.9 °C, salinity from 33.8 to 34.1, and pH from 8.1 to 8.4. Thus, only minor seasonal changes were observed in these parameters, reflecting the tropical marine environment. Nevertheless, the viral community changed dramatically, particularly when comparing June with March and December (Table 1).

A total of 34,685,413 paired-end sequences (31,010,490–40,980,004 reads) were obtained, and after trimming a mean of 34,627,714 reads was assembled (Appendix A). After quality checks using CheckV and read mapping, taxonomic profiling revealed a total of 17,186 double-stranded DNA (dsDNA) virus contigs, of which 13,176 and 4010 were associated with bacteriophages and eukaryotic viruses, respectively (Appendix A). The common viral contigs with a relative abundance > 0.1% in at least one sample were displayed. Accordingly, 1664 and 1725 viral contigs were identified in March and December, respectively, and 621 viral contigs were detected in June. The viral DNA community was classified using cluster analysis based on Bray–Curtis dissimilarity (Figure 2).

In the seawater sample collected in June, bacteriophages, particularly cyanophages that infect cyanobacteria, increased rapidly and comprised the families *Myoviridae* (41.3%, including the new classification families *Straboviridae*, *Kyanoviridae*, and *Autographiviridae*, and the unclassified family *Caudoviricetes*) and *Siphoviridae* (17.55%, including the unclassified family *Caudoviricetes*). In particular, cyanophages, including those in the genera *Synechococcus* and *Prochlorococcus*, accounted for 34.71% of the total relative abundance of viruses in June, which was higher than that observed in March and December (Figure 3). Seawater samples obtained in March and December comprised viruses that infect eukaryotes, particularly NCLDVs in the families *Mimiviridae* (6.20%), *Phycodnaviridae* (3.20%), and *Pandoraviridae* (1.12%), as well as viruses in the family *Podoviridae* (25.54%, including the common family *Autographiviridae* and unclassified family *Caudoviricetes*). Cyanophages accounted for 21.73% of the viral contigs in March and December. Thus, phages associated with potential prokaryotic cyanobacterial hosts were present at higher levels in June, whereas those associated with potential eukaryotic hosts were present at higher levels in March and December.

The alpha-diversity index was used to assess differences in the DNA viral community composition among the samples collected in different months. The Shannon diversity index in March and December was 6.09 and 6.05, respectively, whereas that in June was 5.53. Similarly, the Chao1 richness index and the relative abundance of DNA viruses exhibited patterns similar to those observed for the Shannon diversity index. The Gini–Simpson evenness index was similar among the samples collected in all three months, ranging between 0.986 and 0.989 (Table 1). The resulting Venn diagrams revealed 1459, 826, and 1442 total viral contigs in March, June, and December, respectively, with 556 viral contigs (29.0% of the total) overlapping among all three sampling times (Figure 4). The viral contigs identified in the March and December samples overlapped 57.1% (1095 contigs), whereas the total viral contigs (110 contigs, 5.7%) in June showed the least distinct appearance compared to the other months. In total, there were 147 (34.8%) eukaryotic virus contigs and 407 (27.3%) bacteriophage contigs, including cyanophages, common to all of the samples. In June, similar to the results for the total viral contigs, eukaryotic viruses and bacteriophages exhibited the highest overlap with the other months. 

Common viral contigs, including those of bacteriophages and eukaryotic viruses, were selected at the species level. In total, 170 viral contigs were used to identify 125 taxa of bacteriophages and 45 taxa of eukaryotic viruses (Figure 5 and Appendix A). Among these taxa, three *Podoviridae* (*Puniceispirillum* phage HMO-2011, and *Pelagibacter* phages HTVC010P and HTVC019P); four *Myoviridae* (*Pelagibacter* phage HTVC008M, P-TIM68, and *Synechococcus* phages S-SM2 and S-SKS1); and one *Sipoviridae* (Cyanophage MED4-117) were identified (Figure 5). *Synechococcus*-associated phages (S-SM2 and S-SKS1) were highly prevalent in June, and *Pelagibacter*-associated phages (HTVC010P, HTVC008M, and HTVC019P) were highly prevalent in March and December. Moreover, five *Mimiviridae* (*Acanthamoeba polyphaga moumouvirus*, *Megavirus chiliensis*, *Cafeteria roenbergensis virus* BV-PW1, *Aureococcus anophagefferens virus*, and *Mimivirus terra*2) increased during these two months compared to June.

The large dynamic change observed in June was attributed to an increase in *Myoviridae* and *Podoviridae* contigs among the bacteriophages at the family level. In particular, the rapid increase in the number of cyanophages, including *Synechococcus* and *Prochlorococcus* phages, may have been due to cyanobacterial infection. In our previous study, Chuuk coastal waters examined in August were dominated by cyanobacteria (41%) [28]. Consistent with our results, Carlson et al. [29] reported that cyanobacteria appeared in high abundance in June in the equatorial ocean, and cyanophages were the dominant organisms in the subtropical ocean. In addition, most of the viral DNA were bacteriophages, which reflects their high relative abundance and can be used to estimate the diversity of bacterial hosts. Among the bacteriophages, those in the family *Myoviridae* were the most common, followed by the families *Siphoviridae* and *Podoviridae* [29]. In particular, members of the *Myoviridae* family are widely and abundantly present in the ocean, and while their characteristics are not well known, they have been reported to infect *Synechococcus* [30]. Thus, the viral composition of our samples is similar to that reported in other marine viral metagenomics studies [31]. These results suggest that bacteriophages potentially affect primary production by changing the structure of prokaryotic communities, including those of cyanobacteria.

We also identified members of the following families in our samples: *Mimiviridae*, *Phycodnaviridae*, *Poxviridae*, *Pandoravirieae*, *Herpesviridae*, and *Iridoviridae*. *Mimiviridae* and *Phycodnaviridae* family members infect numerous eukaryotic plankton species, including haptophytes, chlorophytes, and pelagophytes [32,33]. In particular, NCLDVs, including those in the *Mimiviridae* and *Phycodnaviridae* families, are frequently found in cold regions, such as the Arctic [34,35]. Kim et al. [18] showed that members of the *Mimiviridae* were significantly correlated with *Aureococcus anophagefferens* in the subarctic Kongsfjorden Sea. Thus, our identification of taxa in the *Mimiviridae* family (*Megavirus chiliensis*, *Mimivirus terra*2, *Aureococcus anophagefferens virus*, and *Chrysochromulina ericina* virus) and the *Phycodnaviridae* family (*Cafeteria roenbergensis* virus BV-PW1 and *Orpheovirus* IHUMI-LCC2) in the marine environment of Chuuk State suggests that these viruses may pose a potential threat to marine ecosystems. Philippe et al. [36] reported that Pandoraviruses (family *Pandoravirieae*), which are highly evolved from phycodnaviruses, mainly infect amoeba Legendre et al. [37], and Kang et al. [38] reported that *Pandoraviruses* may be important regulatory factors in *Akashiwo sanguinea* (Dinophyceae) blooms. Thus, pandoraviruses, phycodnaviruses, and mimiviruses may be key to controlling most eukaryotic phytoplankton communities. Poxviruses (*Poxviridae*) infect marine mammals [39], and although *Herpesviridae* and *Iridoviridae* appear to be minor groups, herpesviruses can induce diseases in animals, including humans, and *iridoviruses* can infect marine invertebrates (insects and molluscs) and vertebrates (including at least 20 species of fish).

The diversity and abundance of viruses identified in our study may have been affected by the sampling time; certain species appeared to be present in specific months, and several common species appeared to drive changes in species composition. In particular, the abundance of the ubiquitous bacteriophages changed during the study period. *Pelagibacter* phage HTVC010P was the most common bacteriophage identified in our study. Zhao et al. [40] reported that the most abundant viruses in the Pacific Ocean were *Pelagibacter* phages, in particular *Pelagibacter* HTVC010P. In our previous study [28], we found that *Pelagibacter* was the most common genera of bacteria in Chuuk coastal waters, thus explaining the high prevalence of *Pelagibacter* phages identified in the Chuuk coastal waters in this study. Moreover, *Cellulophaga*, *Pseudomonas*, and *Vibrio* phages appeared to be common phages in the current study. Gong et al. [41] reported that certain phages act as potential “markers” for specific bacteria, including *Cellulophaga*, *Pseudomonas*, and *Vibrio* species, reflecting their frequent representation in Prydz and Chile Bays. Thus, bacteriophages are bacterial pathogens that contribute to the growth and decline of seasonal blooms. *Vibrio* and *Pseudomonas* are Gammaproteobacteria that appear frequently in waters with relatively high organic matter and/or in polluted regions [42]. Sewage treatment facilities in Chuuk State pose a potential and/or ongoing threat to marine ecosystems. Moreover, Chuuk State is not well equipped with sewage treatment facilities, for various reasons, such as limited technological and financial resources. Thus, polluted water flowing into the sea may be a source of contamination. People living in Chuuk State depend on the sea for a large percentage of their food, and the region has a developing fishing industry. Thus, when contaminated water flows into the ocean in an unpurified state, it causes changes in the microbial ecosystem, and pathogenic bacteria may emerge. Accordingly, there is a high possibility of phages infecting these pathogenic bacteria. Therefore, it is necessary to continue monitoring and investigating marine pathogens, including viruses and bacteria, in this region.

## 4. Conclusions

This study is the first metagenomics analysis of the viral communities in the marine region of Chuuk State, providing detailed descriptions of the viral communities during different months or seasons. These data will be useful for comparative analyses with other marine viral communities and for the ongoing monitoring of Chuuk State coastal waters. In ocean ecosystems, a high genetic diversity of viruses is found, even within closely related viruses that infect both heterotrophic and autotrophic prokaryotes and eukaryotes [43,44]. In particular, the results of our previous study [28], when compared with the virus community identified in the present study, revealed that the relationship between viruses and their hosts was closely related. Although the study of DNA viruses in trophic marine ecosystems differs considerably from that of laboratory-based viral infections, isolation assays provide important insights into intra-relationships in ecosystems. Therefore, we believe that the emergence of any viral community influences host distribution depending on the infectious activity. Furthermore, we suggest that viral community structure and infectious activity reflect host distribution [45]. Thus, the continued monitoring of viruses and bacteria in ocean waters is essential to protect specific ecosystems as well as the organisms, including humans, that are dependent on the resources in these waters.

## Figures and Tables

**Figure 1 viruses-15-01293-f001:**
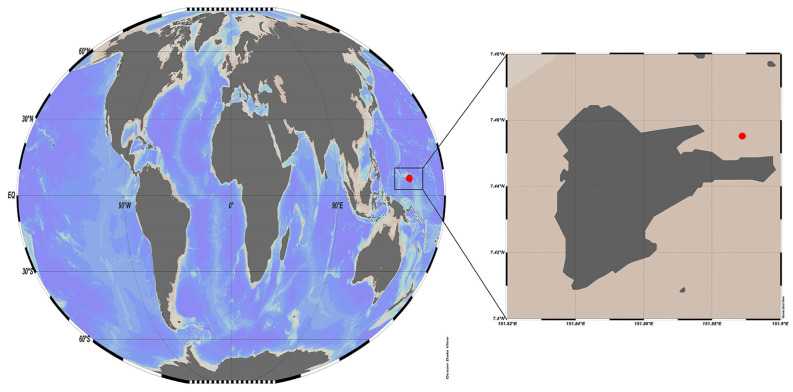
Map of the sampling site. The red circle (7°45.32″ N and 151°89.25″ E) in the red square shows the location of the sampling site in Chuuk coastal waters of the Federated States of Micronesia (FSM).

**Figure 2 viruses-15-01293-f002:**
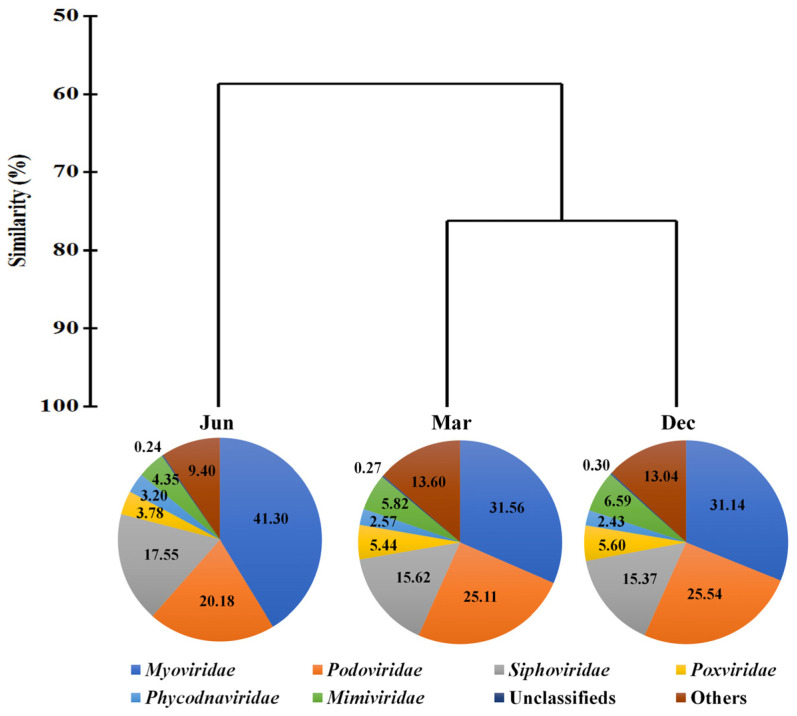
Hierarchical agglomerative clustering performed using the group average of the viral contigs and the Bray–Curtis dissimilarity method. All data were normalised using square root transformation. The pie charts indicate the high-ranking taxonomic distribution at the family level for the viral community. ‘Unclassified’ includes unclassified phages and nucleo-cytoplasmic large DNA viruses (NCLDVs). Unit of pie chart: %.

**Figure 3 viruses-15-01293-f003:**
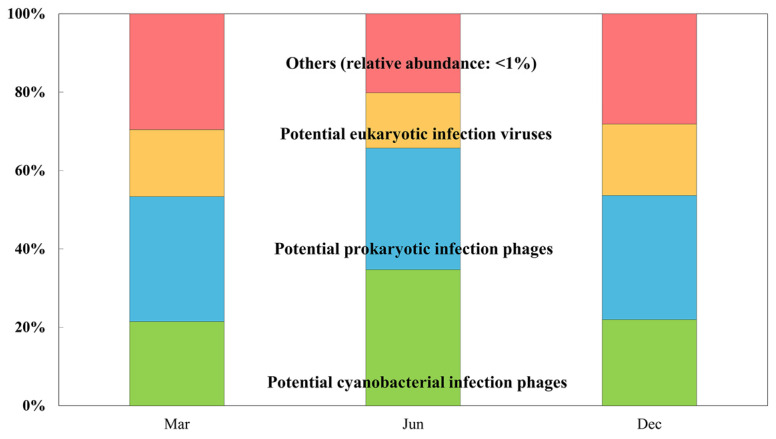
Changes in potential viral infection groups (eukaryotic infection viruses, prokaryotic infection phages, and cyanobacterial infection phages) in Chuuk coastal waters of the FSM. The ‘others’ designation refers to those viruses with a mean relative abundance < 1%.

**Figure 4 viruses-15-01293-f004:**
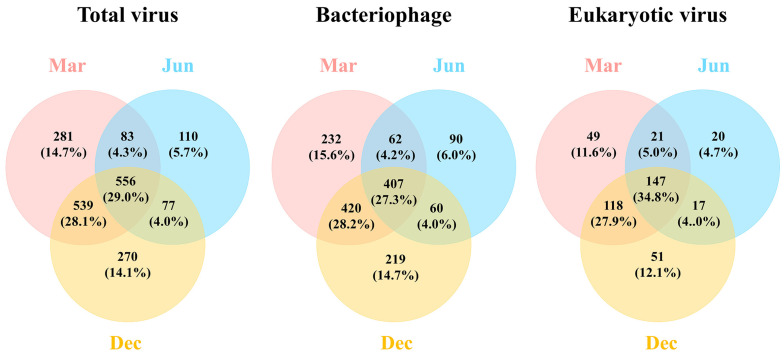
Venn diagram showing the shared and unique common DNA viruses among the three sampling times.

**Figure 5 viruses-15-01293-f005:**
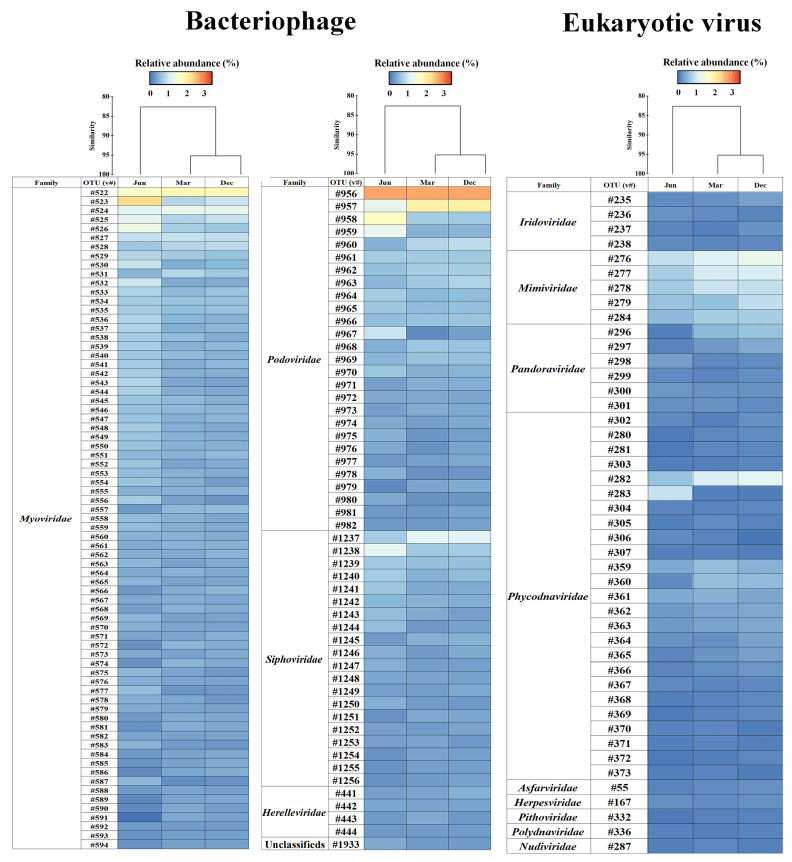
Changes in common DNA viruses (at mean relative abundances > 0.1%) in Chuuk coastal waters of the FSM. The heatmap displays the square root normalised data, ranging from 0 to 3.

**Table 1 viruses-15-01293-t001:** Measured environmental parameters and analysed alpha diversity (Shannon, Chao 1, and Gini–Simpson index).

Sampling Month	Environmental Factors				Alpha Diversity		
	Water Temperature (°C)	Salinity	pH	Fluorescence Chl-*a*	Shannon Diversity	Chao 1 Richness	Gini–Simpson
March	28.6	34.1	8.3	0.31	6.089	2096.03	0.9888
June	29.4	33.8	8.4	0.38	5.563	1662.10	0.9858
December	28.6	33.8	8.1	0.34	6.045	2173.38	0.9887

## Data Availability

The datasets presented in this study can be found in online repositories. The names of the repository/repositories and accession numbers can be found below: https://www.ncbi.nlm.nih.gov/sra/PRJNA972257 (accessed on 18 May 2023).

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
