# Peer review of "Metavirome Profiling and Dynamics of the DNA Viral Community in Seawater in Chuuk State, Federated States of Micronesia"

_viruses, 2023, doi:10.3390/v15061293_

Round 1

Reviewer 1 Report

Comments and Suggestions for Authors

This study described the seasonal changes of phage diversity in the surface water of Chuuk State, Federated States of Micronesia. Based on the NGS results, the authors suggested the seasonal succession of phages would not be affected by surrounding environment. I suggest the authors should provide more environmental parameters and biological parameters (e.g., the abundances of cyanobacteria and bacteria) to  support their conclusion. 

Author Response

RC 1-1: This study described the seasonal changes of phage diversity in the surface water of Chuuk State, Federated States of Micronesia. Based on the NGS results, the authors suggested the seasonal succession of phages would not be affected by surrounding environment. I suggest the authors should provide more environmental parameters and biological parameters (e.g., the abundances of cyanobacteria and bacteria) to support their conclusion.

Response: Thank you for your insightful comment. We agree that analysing the potential host of the virus is necessary. However, quantitative and qualitative analyses of Cyanobacteria and Eubacteria were not performed in this study. Therefore, we tried to perform the analysis using the satellite sea colour data, but it was not included in the results because the sea colour resolution of the chlorophyll data was poor (see figure below). Therefore, based on the 2013 data investigated by my research team (see the reference to Suh et al. 2014 in the manuscript), the ecological relationship between the viruses and their hosts was explained.

We have added the sentence “In particular, the results of our previous study [28], when compared with the virus community identified in the present study, revealed that the relationship between viruses and their hosts was closely related.” in lines 289–291 of the revised manuscript according to the reviewer’s comment.

Reviewer 2 Report

Comments and Suggestions for Authors

The authors describe the variation in the types of dsDNA viruses found in seawater collected at 3 times from a specific location in Micronesia (Chuuk State). The research is solid and contributes to the ongoing efforts to try and understand the variety and ecology of viruses that are present in marine environments. I only have a couple of minor comments.

1. Fig. 2 is very difficult to read. I would move the names to the side and include the colors with them. 

2. line 163. Remove period after December.

3. line 281. Change may to will.

Author Response

Please check the an attached PDF file

Round 2

Reviewer 1 Report

Comments and Suggestions for Authors

I recommend this revised manuscript could be considered to be published in this journal.